# OpenReview forum: "LVLM-COUNT: Enhancing the Counting Ability of Large Vision-Language Models"
_ICLR.cc/2025/Conference — Submitted to ICLR 2025_

### Official Review · Reviewer_wTin · 2024-10-27

**Soundness:** 3
**Presentation:** 2
**Contribution:** 3
**Rating:** 6
**Confidence:** 4

**Summary:**

This paper proposes LVLM-Count, a plug-in method for current LVLM to enhance the visual perception on counting ability. The method uses an extra module with object-aware division, which can divide multi objects without cutting through objects of interest. Also the paper proposes a dataset for testing the counting ability of LVLMs. Compering other method, the proposed method achieves good performance on different benchmarks.

**Strengths:**

This paper provides good thinking and novel insights for a very important domain about LVLMs: visual counting ability of LVLM.

* The paper proposes a novel and simple method for dividing objects without cutting, which is meaningful for visual perception and the inference logic of LVLMs.

* The paper can deal with complex scenarios and achieve high performance on large mount of objects.

* The proposed benchmark looks very interesting.

**Weaknesses:**

I have some concerns about the paper and hope the authors can address them:

* The motivation and method looks good but the experiment results may not support them very well. In the section of experiments (Section 4), there should be some ablations about the designed method.

* The metrics of evaluation may not be enough for well evaluating a LVLM's counting ability. Maybe there are two situations about counting: 1) we need an approximate number of objects. 2) we need an exact number of objects. Current metrics may can only evaluate the first case. So I think you could add more metrics on all benchmarks, such as Accuracy (right cases / total cases, same with EA, but EA is only used for one benchmark). Also you can set different ranges of the Acc, e.g., you can firstly set Acc_{+-0} which means that the answer must be the exact number of objects without any difference. Then you can set Acc_{+-3} which means that the answer is acceptable with in +-3 error numbers. Following this you can set Acc_{+-5}, Acc_{+-10}.

* The method you proposed is actually a plug-in method for any LMLM. So it is a little weak that you only combine it with one LVLM (GPT4o). You should involve more LVLMs such as LLaVA series, Genimi, etc.

* The outputs of the LVLM are always natural languages, how do you make sure that every response is about the counting? how do you avoid the problem of the model giving irrelevant responses?

* In addition to the very incomplete quantitative results, there are also very few qualitative results. The qualitative results are only tested on one dataset. I would like to see the qualitative results and accuracy of image_00001.png, image_00005.png, image_00013.png, image_00036.png, image_00068.png in emoji_benchmark.

The main problem is the incompleteness of the experimental part. If the authors have a positive response, I will consider raising my rating.

**Questions:**

The main evaluations have been listed. I hope the authors can think about the following questions:

* How can the counting ability of LVLMs be better utilized?

* As LVLMs become more and more powerful, will the method proposed in the paper soon become outdated?

* The proposed method seems to work well when there are a large number of objects, so what is its upper limit? Is it okay when there are 500 objects? How about 1000 objects? So what are the advantages of the proposed method?

---

> ### Author Response · Authors · 2024-11-30
> **Response to Reviewer wTin (1/5)**
>
> We sincerely appreciate  reviewers  positive views about the strengths  of our paper and all the concerns that they have raised to help us improve the quality of our work. Below, we address the reviewer’s questions and concerns one by one. Please let us know if you have additional questions or concerns; we are happy to discuss them further.
>
>
>
> Following your highly beneficial and constructive comments, we have made significant efforts to strengthen the experimental aspects of the paper. Accordingly, in the revised manuscript, we have included additional  baselines, experiments on new datasets, ablation studies, and visual examples illustrating the performance of LVLM-Count on various benchmarks and real-world applications.
>
>
>
> For quick access to revisions addressing your specific comments, we have highlighted them in "**orange**”. Additionally, the figure and table numbers in our responses correspond to those in the revised manuscript. For your convenience, we have embedded anonymized links in the titles of each figure and table mentioned in our replies, allowing you to view them by simply clicking the title.
>
>
>
> > The motivation and method looks good but the experiment results may not support them very well. In the section of experiments (Section 4), there should be some ablations about the designed method.
>
>
>
>  We added two baselines to each benchmark in the manuscript (Tables [1](https://anonymous.4open.science/r/lvlm-count-paper-65B1/tables/table_1.jpg), [2](https://anonymous.4open.science/r/lvlm-count-paper-65B1/tables/table_2.jpg), and [3](https://anonymous.4open.science/r/lvlm-count-paper-65B1/tables/table_3.jpg); please click the link to view the tables) to provide a stronger basis for fair comparison:
>
>
>
> 1.  **"Number of the target segmentation masks"**: In this experiment, after segmenting the targets, we count the number of segmentation masks and treat it as the number of objects of interest.
>
>
>
> 2.  **"Base GPT-4o counting the target segmentation masks"**: In this experiment, after the target segmentation, we provide the segmentation masks as input to GPT-4o and ask it to count the masks.
>
>
>
> In addition, we ran our pipeline with two more LVLMs. Details are provided in our response to Comment 3.
>
>
>
> For ablation studies, Appendix A of the submitted manuscript includes a dedicated Ablation section that presents the results of our ablation studies. These include the effects of each stage, such as area detection, object-aware division (including target segmentation), and their comparison with the base GPT-4o and our full pipeline, LVLM-Count.
>
>
>
> In the revised version, we have added further ablations to [Table 4](https://anonymous.4open.science/r/lvlm-count-paper-65B1/tables/table_4.jpg) (please click to view) in Appendix A, including:
>
>
>
> -   Assessing the effectiveness of naïve division compared to our object-aware division, with and without area detection.
>
> -   Changing the LVLM to Gemini and repeating the assessment of each stage of our pipeline for Gemini.
>
>
>
> We sincerely appreciate this comment, which motivated us to provide additional experiments to better support our contributions.

---

> ### Author Response · Authors · 2024-11-30
> **Response to Reviewer wTin (2/5)**
>
> >The metrics of evaluation may not be enough for well evaluating a LVLM's counting ability. Maybe there are two situations about counting: 1) we need an approximate number of objects. 2) we need an exact number of objects. Current metrics may can only evaluate the first case. So I think you could add more metrics on all benchmarks, such as Accuracy (right cases / total cases, same with EA, but EA is only used for one benchmark). Also you can set different ranges of the Acc, e.g., you can firstly set Acc_{+-0} which means that the answer must be the exact number of objects without any difference. Then you can set Acc_{+-3} which means that the answer is acceptable with in +-3 error numbers. Following this you can set Acc_{+-5}, Acc_{+-10}.
>
>
>
> Thank you for your suggestion. It is an excellent idea to provide exact accuracy (EA), Acc{±1}, Acc{±3}, Acc{±5}, and Acc{±10}, along with other performance metrics, to offer deeper insights into the performance of the methods across different accuracy levels.
>
>
>
> Due to space limitations in the main manuscript, we have added Tables [9](https://anonymous.4open.science/r/lvlm-count-paper-65B1/tables/table_9.jpg), [10](https://anonymous.4open.science/r/lvlm-count-paper-65B1/tables/table_10.jpg), [11](https://anonymous.4open.science/r/lvlm-count-paper-65B1/tables/table_11.jpg), and [12](https://anonymous.4open.science/r/lvlm-count-paper-65B1/tables/table_12.jpg)  (please click to view) in Appendix L, which provide all the requested accuracies for each of the result tables (Tables [1](https://anonymous.4open.science/r/lvlm-count-paper-65B1/tables/table_1.jpg), [2](https://anonymous.4open.science/r/lvlm-count-paper-65B1/tables/table_2.jpg), and [3](https://anonymous.4open.science/r/lvlm-count-paper-65B1/tables/table_3.jpg)). We greatly appreciate this suggestion.
>
>
>
> Below is a summary of our observations:
>
>
>
> i) **FSC-147 ([Tables 9](https://anonymous.4open.science/r/lvlm-count-paper-65B1/tables/table_9.jpg))**: For this dataset, similar to the results in Table 1, LVLM-Count achieves higher accuracy metrics compared to prior training-free method and the base LVLMs it employs. However, in general, models specifically trained on this dataset exhibit higher accuracies.
>
>
>
> ii) **TallyQA Simple Benchmark ([Table 10](https://anonymous.4open.science/r/lvlm-count-paper-65B1/tables/table_10.jpg))**: For this benchmark, LVLM-Count outperforms all prior counting models in terms of accuracy. However, it does not achieve higher accuracy compared to the base LVLMs. This is because the questions are simple, and the ground truth values are restricted to a range of 0 to 15—a range where the base LVLMs excel. This aligns with the observations in [Figure 2](https://anonymous.4open.science/r/lvlm-count-paper-65B1/Images/figure_2.jpg) and [Figure 18](https://anonymous.4open.science/r/lvlm-count-paper-65B1/Images/figure_18.jpg) (please click to view).
>
>
>
> iii) **TallyQA Complex Benchmark ([Table 11](https://anonymous.4open.science/r/lvlm-count-paper-65B1/tables/table_11.jpg))**: For this benchmark, LVLM-Count achieves significantly higher accuracy compared to prior counting models. More importantly, it consistently shows accuracy improvements over the base LVLMs.
>
>
>
> iv) **Emoji-Count ([Table 12](https://anonymous.4open.science/r/lvlm-count-paper-65B1/tables/table_12.jpg))**: This is a challenging benchmark with high object counts. As a result, LVLM-Count generally achieves substantially higher accuracies compared to both base LVLMs and prior counting models, with a significantly large margin, especially for Acc{±k} where k≥1

---

> ### Author Response · Authors · 2024-11-30
> **Response to Reviewe wTin (3/5)**
>
> >The method you proposed is actually a plug-in method for any LMLM. So it is a little weak that you only combine it with one LVLM (GPT4o). You should involve more LVLMs such as LLaVA series, Genimi, etc.
>
>
>
> Following your suggestion in this comment, we addressed one of the main shortcomings of the current manuscript—demonstrating LVLM-Count's robustness to changes in the underlying LVLM and its effectiveness. We are grateful for this suggestion.
>
>
>
> We ran all benchmarks in the manuscript (Tables [1](https://anonymous.4open.science/r/lvlm-count-paper-65B1/tables/table_1.jpg), [2](https://anonymous.4open.science/r/lvlm-count-paper-65B1/tables/table_2.jpg), and [3](https://anonymous.4open.science/r/lvlm-count-paper-65B1/tables/table_3.jpg); please click to view) on another commercial LVLM, Gemini 1.5 Pro, and an open-source LVLM named Qwen2 VL 72B AWQ Ref[1]. Although Qwen2 VL 72B AWQ is not as powerful as commercial counterparts such as GPT-4o or Gemini 1.5 Pro, it was the state-of-the-art open-source model at the time of our experiments and outperformed models such as the LLaVA series. The general conclusion is that the performance of LVLM-Count is robust to the choice of the LVLM it uses.
>
>
>
> To be more specific, let us discuss the benchmarks one by one:
>
>
>
> **FSC-147, [Table 1](https://anonymous.4open.science/r/lvlm-count-paper-65B1/tables/table_1.jpg):** Using LVLM-Count significantly improves results across all three base LVLMs. Moreover, all three LVLMs outperform the only prior training-free model, TFOC, when integrated with LVLM-Count.
>
>
>
> **Open-Ended Counting Benchmark - Simple, [Table 2](https://anonymous.4open.science/r/lvlm-count-paper-65B1/tables/table_2.jpg):** All three LVLMs with LVLM-Count outperform prior models. Using LVLM-Count also improves the performance of GPT-4o, but it does not enhance Gemini 1.5 Pro or Qwen2 VL 72B AWQ. The reason is that these questions are simple, with numbers in the range [0, 15]. From the example of counting blue circles in Figure 2 and the error analysis of LVLM-Count for each ground truth interval in Figure 18, we know that base LVLMs outperform LVLM-Count for numbers smaller than ~20. However, for larger numbers, LVLM-Count performs better and margin grows rapidly.
>
>
>
> **Open-Ended Counting Benchmark - Complex, Table 2:** LVLM-Count shows consistent improvements over all three LVLMs and all prior models in terms of exact accuracy (EA).
>
>
>
> **Emoji-Count Benchmark, [Table 3](https://anonymous.4open.science/r/lvlm-count-paper-65B1/tables/table_3.jpg):** LVLM-Count outperforms all prior models and all three base LVLMs by a significant margin.
>
>
>
> The most interesting observation from our experiments is that the performance improvement margin achieved by LVLM-Count compared to the base LVLM is much higher for an open-source model such as Qwen2 VL 72B AWQ, which is considerably weaker than commercial counterparts. Despite the significantly weaker counting performance of Qwen2 VL 72B AWQ compared to GPT-4o and Gemini 1.5 Pro, in most cases, the counting ability of Qwen2 VL 72B AWQ after being used in our pipeline can even be considered on par with models such as GPT-4o and Gemini.
>
>
>
> Please note that we also tested our method with all three base LVLMs on a new benchmark, PASCAL VOC. The results and discussion can be found in Appendix K and [Table 7](https://anonymous.4open.science/r/lvlm-count-paper-65B1/tables/table_7.jpg). As expected, LVLM-Count improves the performance of all three base LVLMs on this benchmark as well.

---

> > ### Author Response · Authors · 2024-11-30
> > **Responce to Reviewer wTin (4/5)**
> >
> > >The outputs of the LVLM are always natural languages, how do you make sure that every response is about the counting? how do you avoid the problem of the model giving irrelevant responses?
> >
> >
> >
> > Thank you for paying attention to important details and raising this question. Commercial models such as GPT-4o and Gemini 1.5 Pro have recently added an in-built feature to their APIs that allows users to request structured output by specifying the format of the response. This way, we can simply request an integer as the final answer. For open-source models such as Qwen2 VL 72B AWQ, the basic idea is the same, though it requires more effort. We instruct it to format the required number as **bold** text, and then use a regular expression to find the bold number in the expected format. If the format is not structured as instructed, we query the model again until the desired format is obtained in the output text.
> >
> >
> >
> > >In addition to the very incomplete quantitative results, there are also very few qualitative results. The qualitative results are only tested on one dataset. I would like to see the qualitative results and accuracy of image_00001.png, image_00005.png, image_00013.png, image_00036.png, image_00068.png in emoji_benchmark.
> >
> >
> >
> > We did our best to provide more quantitative results according to the guidance in your comments. Additionally, we have added Section M in the appendix, which provides visual examples from all the benchmarks: FSC-147 in [Figure 23](https://anonymous.4open.science/r/lvlm-count-paper-65B1/Images/figure_23.jpg), TallyQA Simple in [Figure 24](https://anonymous.4open.science/r/lvlm-count-paper-65B1/Images/figure_24.jpg) (please click the link to view), and TallyQA Complex benchmark in [Figure 25](https://anonymous.4open.science/r/lvlm-count-paper-65B1/Images/figure_25.jpg) (please click the link to view). We have also included visual examples of the performance of LVLM-Count on the requested images from the Emoji-Count benchmark in Figures [26](https://anonymous.4open.science/r/lvlm-count-paper-65B1/Images/figure_26.jpg) and [27](https://anonymous.4open.science/r/lvlm-count-paper-65B1/Images/figure_27.jpg) (please click the link to view).
> >
> >
> >
> > Please also note that other sections have been added to the appendix, showing more qualitative examples:
> >
> >
> >
> > -   Figures [15](https://anonymous.4open.science/r/lvlm-count-paper-65B1/Images/figure_15.jpg), [16](https://anonymous.4open.science/r/lvlm-count-paper-65B1/Images/figure_16.jpg), and [17](https://anonymous.4open.science/r/lvlm-count-paper-65B1/Images/figure_17.jpg) (please click to view) in Appendix G show the performance of our method on real-life examples from health, industry, and environmental monitoring, respectively.
> >
> >
> >
> > -   [Figure 14](https://anonymous.4open.science/r/lvlm-count-paper-65B1/Images/figure_14.jpg) (please click the link to view) in Appendix F shows the performance of LVLM-Count when there are multiple object categories in the image, and we are interested in counting instances of a single category or two categories combined.
> >
> >
> >
> >
> >
> > We cannot express enough appreciation for your special attention to specific examples from our newly introduced and challenging Emoji-Count benchmark.

---

> > > ### Author Response · Authors · 2024-11-30
> > > **Response to Reviewer wTin (5/5)**
> > >
> > > ## Questions:
> > >
> > >
> > >
> > > > 1-How can the counting ability of LVLMs be better utilized?
> > >
> > >
> > >
> > > There might be numerous other parallel methods to ours that can be designed to better utilize LVLMs for counting. However, one interesting observation from our initial experiments is that for simple questions such as 'How many birds are there?', if the original background of the image is removed and replaced with a simpler background that is more relevant to the object category, without compromising the quality of the objects of interest, this context helps reduce counting errors. For example, removing the background from an image containing birds and replacing it with a simple blue sky background can reduce the counting error. This could be a future direction for better utilizing LVLMs for simple counting questions.
> > >
> > >
> > >
> > > >2.  As LVLMs become more and more powerful, will the method proposed in the paper soon become outdated?
> > >
> > > >3.  The proposed method seems to work well when there are a large number of objects, so what is its upper limit? Is it okay when there are 500 objects? How about 1000 objects? So what are the advantages of the proposed method?
> > >
> > >
> > >
> > > We answer questions 2 and 3 together, as they are related.
> > >
> > >
> > >
> > > First, let us start with question 3 about the upper limit of our method. The limit depends on:
> > >
> > > how many objects the LVLM can count accurately in a subimage,
> > >
> > > and how many subimages are created during third stage  of our method
> > >
> > >
> > >
> > > Currently, based on our observations, LVLMs such as GPT-4o and Gemini 1.5 Pro perform very well for numbers from 0 to around 20 on average, though this may vary depending on the dataset.
> > >
> > >
> > >
> > > The number of subimages is detected automatically during third stage of our method. Therefore, this number depends on the input image, and it is difficult to determine a-priori. However, we can provide some empirical evidence. For example, for FSC-147 the method splits the images into approximately 9 subimages (each as a patch in a 3x3 grid of sub images). Since each subimage is counted by an LVLM, the result for each subimage will be accurate if it contains up to 20 objects of interest. If each subimage contains up to 20 objects, then, this means our method will perform well for counting up to  9 * 20 = 180 objects. If a subimage contains more than 20 objects, then the accuracy may be lower.
> > >
> > >
> > >
> > > Now, regarding question 2, let us assume that a day will come when state-of-the-art LVLMs can count accurately enough for numbers up to 100. In this case, although LVLM-Count's importance may decrease for numbers below 100, this will increase LVLM-Counts's upper limit to 9 * 100 = 900. Thus, our method will not be obsolete; rather, its relevance will shift from a certain range of ground truth numbers to higher values.
> > >
> > >
> > >
> > > ----------
> > >
> > > At the end, we are deeply grateful for all your constructive comments and suggestions, which helped us alleviate the weaknesses of our manuscript by providing a more extensive set of quantitative and qualitative examples in accordance with your feedback. We sincerely thank the reviewer and welcome any further comments. We hope the clarifications and additional results provided are sufficient for them to reconsider their score.
> > >
> > >
> > >
> > > ---
> > >
> > > References:
> > >
> > > Ref[1]  An Yang, Baosong Yang, Binyuan Hui, Bo Zheng, Bowen Yu, Chang Zhou, Chengpeng Li, Chengyuan Li, Dayiheng Liu, Fei Huang, et al. Qwen2 technical report. arXiv preprint _arXiv:2407.10671, 2024._

---

> ### Comment · Reviewer_wTin · 2024-11-30
>
> Many thanks for the response. I think your response has basically resolved my doubts, so I am willing to raise my score to 6. Some small suggestions: I think the paper could be modified with more deep thinkings about the learning mechanism of LVLMs about counting and the figures could be designed to be more beautiful.

---

> > ### Author Response · Authors · 2024-12-01
> >
> > We would like to express our gratitude to the reviewer for raising their score. We will also take into account your additional comments for our paper. Thank you.

---

### Official Review · Reviewer_4Qwc · 2024-11-03

**Soundness:** 2
**Presentation:** 3
**Contribution:** 2
**Rating:** 3
**Confidence:** 5

**Summary:**

This paper proposes LVLM-Count, a prompt-based, training-free counting approach that enhances object counting performance with large vision-language models (LVLMs).  For addressing limitations with current LVLMs, such as their challenges with high-count tasks and complex counting questions, this method employs a four-stage pipeline: (1) area detection, (2) target segmentation, (3) object-aware division, and (4) target counting. This enables LVLMs to effectively segment, process, and count large numbers of objects across diverse datasets, showcasing robust generalization.

**Strengths:**

A simple and intuitive pipeline for counting with LVLM

Good presentation along with clear drawn figures.

A newly introduced Emoji-Count benchmark is introduced, though the generation of this data is not complex but still useful as a testbed.

Good performance margin achieved.

**Weaknesses:**

W1: At the very beginning, the authors should define more clearly what means by large number of objects, 10s, 100s, or 1000s, per image. As this defines the scope of this work in terms of crowdedness.

W2: The key idea of this work, divide-and-conquer, can be hardly considered novel for two reasons: 1) in this context, counting by definition is a process of adding the number of objects from region to region. It is essential a process of summing up across regions; 2) Such an idea has appeared in the counting literature such as [Ref 1] where there is also a need for avoiding repeatedly computing regions within an image. As a result, this whole method pipeline is not sufficiently novel -- it is more like a baseline design of using LVLM for counting.
- [Ref 1] Xiong H, Lu H, Liu C, Liu L, Cao Z, Shen C. From open set to closed set: Counting objects by spatial divide-and-conquer. InProceedings of the IEEE/CVF international conference on computer vision 2019 (pp. 8362-8371).

W4: It is inconsistent than different examples are used from Fig 3 to Fig 5 when discussing individual components. From these examples, little challenges are visible with counting, and I am impressed that simply counting the mask of GroundingDINO would achieve good performance, even not bother LVLM such as GPT-4o. For example, simply counting the mask number in Fig 4(c) can give us very accurate count. I would suggest the authors use the same example with proper challenges involved and considered in this work, across all these sections.

W5: This method uses a number of pretrained models such as LLMs, GroundingDINO, and GPT-4o, Real-ESRGAN. One concern is about efficiency. The authors should conduct an efficiency analysis in both training and inference, which is now missing.

W6: Except the comparison with previous works, I suggest a couple of baseline methods should be included in this work:
1) LLM + GroundingDINO: After target segmentation step (Sec 3.2), directly counting the masks by SAM. This can be use to validate the significance of region division, a key aspect of this work. This complements to the ablation result of using GPT-4o in Table 4 (Appendix).

2) Passing the SAM's mask to GPT-4o to count: This will directly compare the proposed object aware division algorithm.

3) To compare with Ref 1's strategy on avoiding repeated counting at the region level

W7: In ablation study, it is suggested that the authors examine the effect of using super-resolution on the regions.

W8: Please clarify what LLM is used in this work?

W9: Except those datasets used, another good test is PASCAL VOC. This should add different test cases on top. The authors can consider to evaluate.

**Questions:**

Please check weaknesses above

---

> ### Author Response · Authors · 2024-11-30
> **Response to Reviewer 4Qwc (1/7)**
>
> We sincerely thank the reviewer for their time and insightful feedback on our manuscript. Below, we address the reviewer’s questions and concerns one by one. Please let us know if you have additional questions or concerns; we are happy to discuss them further.
>
>
>
>  We have performed every experiment that you requested. In particular, in the revised manuscript, we have included additional baselines, experiments on new datasets, ablation studies, and visual examples illustrating the performance of **LVLM-Count** on various benchmarks and real-world applications.
>
>
>
> For quick access to revisions addressing your specific comments, we have highlighted them in **brown**. Additionally, the figure and table numbers in our responses correspond to those in the revised manuscript. For your convenience, we have embedded anonymized links in the titles of each figure and table mentioned in our replies, allowing you to view them by simply clicking the title.
>
>
>
> >W1: At the very beginning, the authors should define more clearly what means by large number of objects, 10s, 100s, or 1000s, per image. As this defines the scope of this work in terms of crowdedness.
>
>
>
> We truly appreciate your attention to this important detail. Indeed, this requires clarification in the manuscript to avoid ambiguity for the reader. The range within which current base LVLMs perform very well without requiring our method includes numbers from 0 to approximately 20. We have added this approximate range to line 036 in the revised manuscript.
>
>
>
> Please note that 20 is an approximate average based on our observations of the performance of base LVLMs across different datasets. This trend can be observed, for example, in [Figure 2](https://anonymous.4open.science/r/lvlm-count-paper-65B1/Images/figure_2.jpg) (please click the link to view the figure), where base GPT-4o performs better when counting less than 20 circles compared to cases where the images are divided.
>
>
>
> To further support this observation, we provide an analysis of the performance of base LVLMs and our method (LVLM-Count) across different ground truth ranges in the FSC-147 dataset in Section H of the appendix. The results are presented in [Figure 18](https://anonymous.4open.science/r/lvlm-count-paper-65B1/Images/figure_18.jpg) (please click to view the figure). These diagrams clearly show that for numbers below 20, the base LVLMs exhibit better performance, whereas for larger numbers, our method outperforms the base LVLMs. Moreover, the performance gain increases significantly in our favor as the numbers grow larger.

---

> ### Author Response · Authors · 2024-11-30
> **Response to Reviewer 4Qwc (2/7)**
>
> >W2: The key idea of this work, divide-and-conquer, can be hardly considered novel for two reasons: 1) in this context, counting by definition is a process of adding the number of objects from region to region. It is essential a process of summing up across regions; 2) Such an idea has appeared in the counting literature such as [Ref 1] where there is also a need for avoiding repeatedly computing regions within an image. As a result, this whole method pipeline is not sufficiently novel -- it is more like a baseline design of using LVLM for counting.
>
> >-   [Ref 1] Xiong H, Lu H, Liu C, Liu L, Cao Z, Shen C. From open set to closed set: Counting objects by spatial divide-and-conquer. InProceedings of the IEEE/CVF international conference on computer vision 2019 (pp. 8362-8371).
>
>
>
> ***First part of comment W2**: The key idea of this work, divide-and-conquer, can be hardly considered novel for two reasons: 1) in this context, counting by definition is a process of adding the number of objects from region to region. It is essential a process of summing up across regions;*
>
>
>
> We completely agree with you that no one should claim novelty for the concept of divide and conquer itself. In our paper, we claim novelty in *how we divide the images, not in the divide-and-conquer idea*. Specifically, we introduce an “object-aware” division mechanism, which is the key contribution of our paper. This mechanism allows images to be divided without cutting through instances of a specified (by a prompt) object category, a capability that is entirely novel.
>
>
>
> When this mechanism is absent, sections of an object divided by naïve division lines are counted independently, leading to significantly inflated final answers. This issue is clearly illustrated in the example of blue circles, where the naïve division approach results in a higher error compared to our object-aware division mechanism ([Figure 2](https://anonymous.4open.science/r/lvlm-count-paper-65B1/Images/figure_2.jpg); please click to view the figure).
>
>
>
> To further support this claim, we have added additional experiment to the ablation studies ([Table 4](https://anonymous.4open.science/r/lvlm-count-paper-65B1/tables/table_4.jpg) in Appendix A; please click the link to view the table) in which naïve divisions are used instead of object-aware divisions. These entries in the table are labeled “Naive Division.” It is immediately evident that the performance of any pipeline using naïve division, rather than object-aware division, not only fails to improve over the base LVLM but also exhibits significantly worse performance, with MAE degrading by more than 15 units! This degradation arises from multiple counts of bisected objects.
>
>
>
> In stark contrast, all experiments using the object-aware division mechanism show substantial improvements over the base LVLMs, including both GPT-4o and Gemini 1.5 Pro. This demonstrates the importance and novelty of our contribution.
>
>
>
> To the best of our knowledge, we are the first to propose a solution for object-aware image division and to incorporate it into a pipeline to enhance the counting ability of large vision-language models.

---

> ### Author Response · Authors · 2024-11-30
> **Response to Reviewer 4Qwc (3/7)**
>
> ***Second  part of comment W2**: 2) Such an idea has appeared in the counting literature such as [Ref 1] where there is also a need for avoiding repeatedly computing regions within an image. As a result, this whole method pipeline is not sufficiently novel *
>
>
>
> Thank you for bringing to our attention Ref[1]. We will make sure to cite it. We would like to point out that Ref[1] is a supervised method, which can only count a single category on which it was trained on. Our method is unsupervised, and due to the LVLMs it can do general object counting. We provide comparisons later in response to comment W6 where we show that our method significantly outperforms Ref[1] for general-purpose counting.
>
>
>
> Our paper is the first to illustrate how one can utilize a powerful, general-purpose counter, under the framework of divide and conquer, by avoiding double counting for arbitrary object category without supervision.
>
>
>
> The novelty of our work is the approach that we used (object-aware division) to avoid double counting while at the same using a general-purpose counter.
>
>
>
> > -- it is more like a baseline design of using LVLM for counting.
>
>
>
> A baseline LVLM for counting is the one that does naive division of an image. This is simply dividing an image using straight vertical and horizontal lines.  We don’t do this in our paper. In our paper we have the object-aware division mechanism to avoid double counting. In fact, we have a comparison in our paper to naive division (please see [Table 4](https://anonymous.4open.science/r/lvlm-count-paper-65B1/tables/table_4.jpg) in Appendix A, ablation studies), and we illustrate that our method outperforms it significantly.  Not only we outperform naive division, but more importantly naive division results in worse performance also compared to base LVLM.
>
> Let us now discuss Ref[1] and how their divide-and-conquer approach differs in purpose from ours. Ref[1] trains a dedicated convolutional neural network and applies naive division to the feature maps generated by it. The reason they perform division on the feature maps is to avoid the computational cost of recalculating the feature maps for each sub-image repeatedly. When they mention “avoid repeatedly computing sub-region feature-maps,” they aim to address the issue of efficiency, not the issue of double counting. Specifically, they state that in the supervised setting, one could simply apply divide-and-conquer directly to the image. However, they also note: “This way, however, is likely to blur the image and lead to exponentially increased computation cost and memory consumption when repeatedly extracting the feature map.”
>
> Due to these computational challenges, they demonstrate that performing naive divide-and-conquer on the feature maps achieves comparable performance in the supervised setting while offering computational advantages.
>
>
>
>
>
>
>
> The problem of counting accuracy during image-level divide-and-conquer in the supervised setting was first addressed in Ref[2], which we cite and acknowledge in line 058 of our paper. However, Ref[2] does not solve this issue for arbitrary object categories. Moreover, their approach requires training dedicated convolutional neural networks, which inherently face the same limitation as any other trained model, i.e., limited knowledge.
>
>
>
> We hope the discussion above has clarified and distinguished our contributions in LVLM-Count. Nonetheless, we are more than happy to cite Ref[1] in our work and sincerely appreciate you raising it for consideration. Please see lines 117 and 118 in the Related Works section.

---

> ### Author Response · Authors · 2024-11-30
> **Response to Reviewer 4Qwc (4/7)**
>
> >W4: It is inconsistent than different examples are used from Fig 3 to Fig 5 when discussing individual components. From these examples, little challenges are visible with counting, and I am impressed that simply counting the mask of GroundingDINO would achieve good performance, even not bother LVLM such as GPT-4o. For example, simply counting the mask number in Fig 4\(c\) can give us very accurate count. I would suggest the authors use the same example with proper challenges involved and considered in this work, across all these sections.
>
>
>
> **We believe this comment has two independent parts:**
>
>
>
> #### **Part i: Consistent Use of a Single Image**
>
>
>
> Thank you for raising this point about the presentation of images in the manuscript. We agree that it is an excellent idea to depict the transformations at each stage using a consistent image. Taking your valuable advice, we have added **Section I** to the appendix. In this section, we demonstrate all the steps for the same zebra image used in [Figure 1](https://anonymous.4open.science/r/lvlm-count-paper-65B1/Images/figure_1.jpg) of the main manuscript, which illustrates our pipeline. **Figures [19](https://anonymous.4open.science/r/lvlm-count-paper-65B1/Images/figure_19.jpg), [20](https://anonymous.4open.science/r/lvlm-count-paper-65B1/Images/figure_20.jpg), and [21](https://anonymous.4open.science/r/lvlm-count-paper-65B1/Images/figure_21.jpg)** (linked for reference) in the appendix now show these steps for the zebra image, highlighting the challenges encountered at each stage.
>
>
>
> Please note that we intentionally used simple and illustrative examples in the main text to ensure clarity and ease of understanding for readers. However, thanks to your feedback, Section I now provides a comprehensive demonstration of what happens to a single image as it progresses through all stages of our method. This addition complements the simpler examples in the main text, allowing readers to gain a deeper understanding of the challenges involved.
>
>
>
> ----------
>
>
>
> #### **Part ii: Masks in Figure 4**
>
>
>
> You raised a valid concern about the potential redundancy of using an LVLM instead of simply relying on masks, specifically referencing **[Figure 4](https://anonymous.4open.science/r/lvlm-count-paper-65B1/Images/figure_4.jpg)** in the manuscript (linked for reference). In fact, Figure 4 was deliberately chosen to demonstrate why counting masks alone is insufficient. For example, the expression *E* in Figure 4 specifies "***brown*** eggs," but as shown, all eggs—irrespective of color—are segmented. This mismatch leads to significant errors if one were to count masks directly. In fact, this incident is very common for complex questions that contain a referring expression such as Emoji-count ([Table 3](https://anonymous.4open.science/r/lvlm-count-paper-65B1/tables/table_3.jpg)) and Open-ended counting benchmark ([Table 2](https://anonymous.4open.science/r/lvlm-count-paper-65B1/tables/table_2.jpg)).
>
>
>
> To have a visual example in the manuscript that clarifies, we have added **Section J** to the appendix, along with **[Figure 22](https://anonymous.4open.science/r/lvlm-count-paper-65B1/Images/figure_22.jpg)** (linked for reference), which provides clear examples where counting masks results in large errors. These examples explicitly illustrate that false-positive masks do not negatively impact LVLM-Count's final predictions; their only effect is on the division lines, which will not segment false-positive objects too. We sincerely appreciate your comment, as it helped us clarify and strengthen this aspect of our manuscript.

---

> ### Author Response · Authors · 2024-11-30
> **Response to Reviewer 4Qwc (5/7)**
>
> >W5: This method uses a number of pretrained models such as LLMs, GroundingDINO, and GPT-4o, Real-ESRGAN. One concern is about efficiency. The authors should conduct an efficiency analysis in both training and inference, which is now missing.
>
>
>
> Thank you for raising the concern about the efficiency analysis. The following table shows the approximate training and inference times of different methods:
>
> |                                              | Training (h)  | Inference (s)  |
> |----------------------------------------------|:-------------:|:--------------:|
> | TFOC                                         |       0       |      0.83      |
> | DAVEprm                                      |      ~36      |      0.27      |
> | GroundingREC                                 |      ~11      |      0.09      |
> | CountGD                                      |      ~24      |      0.17      |
> | GPT4o                                        |       -       |      1.68      |
> | Ours (LVLM-Count, sequential)  |       0       |      6.26      |
> | Ours (LVLM-Count, parallel)    |       0       |      2.82      |
>
> Please note that our method does not require training and therefore has a training time of 0.
>
>
>
> Furthermore, The table below provides a breakdown of the inference time for sequential and parallel LVLM-Count:
>
> |                                            |    Extract E    |     Area detection     |  Target Segmentation  | Target Segmentation | subimage counting    |   Super resolution   | total   |
> |--------------------------------------------|:---------------:|:----------------------:|:---------------------:|:-------------------:|:--------------------:|:--------------------:|---------|
> |                                            | Call to GPT-4o  | Call to GroundingDINO  | Call to GroundingDIN  | Call to SAM         | Call to GPT-4o       | Call to Real-ESRGAN  | -       |
> | LVLM-Count (sequential)   |           0.56  |                  0.10  |                 0.10  |               0.33  |                4.97  |                0.20  | 6.26    |
> | LVLM-Count ( parallel) *   |           0.56  |                  0.10  |                 0.10  |               0.33  |                1.65  |                0.06  | 2.82    |
> |
>
> \* This value is estimated, see comments below.
>
> The inference time of our code is not optimized, as inference is performed on each sub-image serially. Please note that the majority of the inference time is due to the call to GPT-4o at the end for each sub-image (4.97). However, the call to GPT-4o for each sub-image can be trivially parallelized, as the calls are independent of each other and can be sent simultaneously. In this case, 4.97 seconds is expected to drop to 1.65 seconds, reducing the overall inference time from 6.26 to 2.82 seconds.
>
> To obtain the inference time for each stage, we evaluated the method on 1,000 images from FSC-147  dataset and calculated the average. Each image was divided into three subimages. Please note that since GPT-4o cannot be run locally, a considerable time spent on the queries might be related to net protocols. Thus, it is out of our control.

---

> ### Author Response · Authors · 2024-11-30
> **Response to Reviewer 4Qwc (6/7)**
>
> >W6: Except the comparison with previous works, I suggest a couple of baseline methods should be included in this work:
>
> > LLM + GroundingDINO: After target segmentation step (Sec 3.2), directly counting the masks by SAM. This can be use to validate the significance of region division, a key aspect of this work. This complements to the ablation result of using GPT-4o in Table 4 (Appendix).
> > Passing the SAM's mask to GPT-4o to count: This will directly compare the proposed object aware division algorithm.
> >To compare with Ref 1's strategy on avoiding repeated counting at the region level
>
>
>
> We welcome this excellent suggestion. We experimented with baselines 1 and 2 across all three benchmarks in the manuscript and have added the results to the corresponding tables (Tables [1](https://anonymous.4open.science/r/lvlm-count-paper-65B1/tables/table_1.jpg), [2](https://anonymous.4open.science/r/lvlm-count-paper-65B1/tables/table_2.jpg), and [3](https://anonymous.4open.science/r/lvlm-count-paper-65B1/tables/table_3.jpg); please click on the link to view). The name “Number of the target segmentation masks” represents baseline 1, while “Base GPT-4o counting the target segmentation masks” represents baseline 2. A brief conclusion from these experiments is that LVLM-Count outperforms both baselines by a significant margin, which further emphasizes our contributions. This valuable suggestion helped us better present our method's impact and the substantial improvement it brings over the baselines.
>
>
>
>  Regarding baseline 3, it cannot be applied in our pipeline for avoiding repeated counting. As discussed in detail in response to the second part of comment W2, the strategy in Ref[1] is straightforward divide and conquer on the feature-maps computed by a CNN for an input image and addresses the issue of efficiency, not the issue of error caused by counting a bisected object multiple times. More importantly, their divide and conquer strategy requires training a dedicated convolutional neural network. In other words their strategy requires access to convolutional features as they apply divide and conquer on feature maps. However, we do not have access to an LVLM’s feature maps. Also, note that as soon as we train a dedicated network for counting, we will face the issue of all other trained counting models, i.e. limited knowledge.
>
>
>
> However, we ran S-DCNet on all the benchmarks in the paper. Please note that S-DCNet is only trainable to count one object category. Therefore, our method naturally achieves a much higher score. Please note that these results have not been included in the manuscript, as comparing prompt-based counting methods with a CNN designed specifically to count a single object category is not useful due to the above limitation of S-DCNet. Having this mind, we describe the details of the experiment below.
>
>
>
> In our experiment we had to pick a single category, since there is “person” category in most of the images of the benchmarks that we report on, we use an S-DCNet trained to count humans. In other words, we simply provided an image (without any prompts), and it returned a number corresponding to its estimation of the number of humans in the image. The results are as follows:
>
>
> **FSC-147:**
>
> |                              |  MAE   |  RMSE   |
> |------------------------------|:------:|:-------:|
> | S-DCNet                      | 45.28  | 142.60  |
> | LVLM-Count (GPT-4o as LVLM)  | **16.23**  | **76.09**   |
> ---
> **Open-ended counting benchamrk - simple**
>
> |                              |   EA    |  MAE  |  RMSE
> |------------------------------|:-------:|:-----:|:------:|
> | S-DCNet                      | 7.10%   | 8.51  | 20.88  |
> | LVLM-Count (GPT-4o as LVLM)  | **44.73%**  | **1.18**  | **2.06**
> ----
> **Open-ended counting benchamrk - complex**
>
> |                              |   EA    |  MAE   |  RMSE
> |------------------------------|:-------:|:------:|:------:|
> | S-DCNet                      | 4.03%   | 14.15  | 27.77  |
> | LVLM-Count (GPT-4o as LVLM)  | **31.38%**  | **2.51**   | **4.36**
> ---
> **Emoji-Count benchmark**
>
> |                              |  MAE   |  RMSE  |  RMSE
> |------------------------------|:------:|:------:|:------:|
> | S-DCNet                      | 32.45  | 38.36  | 27.77  |
> | LVLM-Count (GPT-4o as LVLM)  | **9.82**   | **19.46**  | **4.36**
> ---
>
>
> Overall, this comment helped us clarify our contributions, and we are truly grateful.

---

> ### Author Response · Authors · 2024-11-30
> **Response to Reviewer 4Qwc (7/7)**
>
> >W7: In ablation study, it is suggested that the authors examine the effect of using super-resolution on the regions.
>
>
>
> Thank you for this suggestion. We have now added an experiment in the ablation studies ([Table 4](https://anonymous.4open.science/r/lvlm-count-paper-65B1/tables/table_4.jpg) in Appendix A; please click the link to view), where super-resolution is activated.
>
>
>
> We observe that the performance is very close to the version of LVLM-Count, which does not use super-resolution, but it is slightly lower. Moreover, we can confirm that super-resolution does not benefit LVLM-Count in either the FSC-147 or Emoji-Count experiments. Therefore, we did not use it in these cases. The only benchmark where we had to use it was the Open-Ended Counting benchmark. The reason is that, in the Open-Ended Counting benchmark, after the subimages were generated, they lacked sufficient resolution.
>
>
>
> >W8: Please clarify what LLM is used in this work?
>
>
>
> We apologize for failing to clarify this in the manuscript, which may have caused confusion for the reader. The LLM mentioned is the same as the LVLM used in the pipeline. LVLMs are typically composed of a base LLM equipped with vision components (e.g., GPT-4o  vs. GPT-4). We have clarified this on lines 49–50 of the revised manuscript. We appreciate your attention to such important details.
>
>
>
> >W9: Except those datasets used, another good test is PASCAL VOC. This should add different test cases on top. The authors can consider to evaluate.
>
>
>
> This is a great suggestion, and we happily welcome it. The experiments on the PASCAL VOC dataset are discussed in Section K of the appendix and [Table 7](https://anonymous.4open.science/r/lvlm-count-paper-65B1/tables/table_7.jpg)( please click to view).
>
>
>
> Please note that PASCAL VOC is originally annotated for classification, detection, and segmentation. We made some modifications to produce a counting benchmark from the PASCAL VOC 2007 dataset and ran experiments with LVLM-Count using three different LVLMs: GPT-4o, Gemini 1.5 Pro, and Qwen2 VL 72B AWQ (the latter being the state-of-the-art open-source LVLM at the time of conducting the experiments). The discussion on the procedure for creating the benchmark and the results is included in Section K of the appendix and [Table 7](https://anonymous.4open.science/r/lvlm-count-paper-65B1/tables/table_7.jpg).
>
>
>
> The summary of our observations is that LVLM-Count improves the performance of each of the LVLMs tested. Moreover, it outperforms all prior counting methods except CountGD. Upon further investigation, we noticed that the 20 categories in PASCAL VOC overlap significantly with the object categories in the dataset used to train CountGD.
>
>
>
> Please note that PASCAL VOC includes 20 object categories from everyday scenes. The ground truth counts in PASCAL VOC are heavily biased towards numbers below 10. To address this, we randomly sampled five images for each ground truth number present in the dataset. In some cases, there were not enough images available, so we included all the available images. We generated counting questions that simply asked for the count of a specific category (e.g., cats).
>
>
>
> We thank you for your comment, which prompted us to run experiments on PASCAL VOC and provide more insight into the performance of our method for interested readers.
>
>
>
> ---
>
>
>
> Overall, we sincerely thank you for your criticism and valuable comments, which helped us rectify our shortcomings, clarify our contributions and novelty, and provide additional experiments and insights for readers. We are deeply grateful for your time and welcome any further comments. We hope that the clarifications and additional results provided are sufficient to to consider increasing our score.
>
>
>
> ---
>
> References:
>
> Ref [1] Xiong H, Lu H, Liu C, Liu L, Cao Z, Shen C. From open set to closed set: Counting objects by spatial divide-and-conquer. InProceedings of the IEEE/CVF international conference on computer vision 2019 (pp. 8362-8371).
>
>
>
> Ref[2]  Prithvijit Chattopadhyay, Ramakrishna Vedantam, Ramprasaath R Selvaraju, Dhruv Batra, and Devi Parikh. Counting everyday objects in everyday scenes. In Proceedings of the IEEE Conference on Computer Vision and Pattern Recognition, pp. 1135–1144, 2017.

---

> > ### Comment · Reviewer_4Qwc · 2024-11-30
> > **Discussion I**
> >
> > Thank the authors for detailed response and discussion.
> >
> > Can I ask more questions regarding the ablation in Table 4, Appendix:
> >
> > 1. Why do you randomly sample 4 images from each category in the FSC-147 test set (29 categories) and report the performance on the resulting subset containing 116 samples? This clearly raises issues about randomness and also leads inconsistency with the main results in previous table. It is suggested to use all the images.
> >
> > 2. Super-resolution looks being negative in this case, which is inconsistent with the rest. More discussion is needed for when this is needed and when it is not.
> >
> > 3. Which entry corresponds to the baseline of LVLM + GroundingDINO: After target segmentation step (Sec 3.2), directly counting the masks by SAM?
> >
> > 4. How about inputing the LVLM with an image only keeping the SAM mask areas while removing the rest? What results could be obtained?
> >
> > Overall the current ablation study is not convincing yet, there is a need to be more carefully conducted, taking into account the above comments. Thanks.

---

> ### Author Response · Authors · 2024-12-02
> **Reply to Reviewer 4Qwc Additional  Comments (1/4)**
>
> >Why do you randomly sample 4 images from each category in the FSC-147 test set (29 categories) and report the performance on the resulting subset containing 116 samples? This clearly raises issues about randomness and also leads inconsistency with the main results in previous table. It is suggested to use all the images.
>
> Thank you for your comment.
>
> We would like to clarify that the only reason we randomly sampled 4 images from each object category was to build a smaller subset to manage the API cost of conducting a comprehensive ablation study with state-of-the-art LVLMs such as GPT-4o and Gemini. The additional cost for this particular experiment that the reviewer requested is $200 per trial, which quickly adds up with more trials.
>
> However, your comment is of great importance to us. Therefore, we have decided to run the ablation on the entire FSC-147 dataset despite the significant additional cost.
>
> The table in the ablation section has been updated to [Table 4](https://anonymous.4open.science/r/lvlm-count-paper-65B1/tables_updated/table_4_updated.jpg) (please click on the title to view the updated table). This is based on 1,190 images from the test set of the FSC-147 dataset (entire test set). The results (rankings) are consistent with those obtained on the smaller subset in the previous version.
>
> For the reviewer's convenience, we have included the new ablation table here as well. In summary, there is no major change in the ranking of the different versions tested for ablation. Our method still outperforms all other versions by a significant margin, with or without super-resolution.
>
>
>
> **Ablation Study, [Table 4](https://anonymous.4open.science/r/lvlm-count-paper-65B1/tables_updated/table_4_updated.jpg):**
>
>
>
> | Method | MAE | RMSE |
> |-------------------------------------------------------------------------------|-------|--------|
> | GPT4o | 25.17 | 137.86 |
> | GPT-4o + Naive Division | 29.26 | 79.91 |
> | GPT-4o + Area Detection + Naive Division | 46.82 | 327.97 |
> | GPT-4o + Area Detection | 22.86 | 104.08 |
> | GPT-4o + Object-aware Division | 17.15 | 79.63 |
> | GPT-4o + Area detection + Object-aware Divsion (equiv. to LVLM-Count) w/o SR | **16.23** | **76.09** |
> | GPT-4o + Area Detection + Object-aware Divsion (equiv. to LVLM-Count) w SR | 16.77 | 89.60 |
> | Gemini 1.5 Pro | 25.20 | 108.76 |
> | Gemini 1.5 Pro + Naive Division | 49.20 | 150.35 |
> | Gemini 1.5 Pro + Area Detection + Naive Division | 44.67 | 131.25 |
> | Gemini 1.5 Pro + Area Detection | 37.57 | 129.73 |
> | Gemini 1.5 Pro + Object-aware Division | 19.78 | 88.97 |
> | Gemini 1.5 Pro + Area detection + Object-aware Divsion (equiv. to LVLM-Count) w/o SR | **14.85** | **85.60** |
> | Gemini 1.5 Pro + Area Detection + Object-aware Divsion (equiv. to LVLM-Count) w SR | 16.95 | 99.70 |

---

> ### Author Response · Authors · 2024-12-02
> **Reply to Reviewer 4Qwc Additional  Comments (2/4)**
>
> > Super-resolution looks being negative in this case, which is inconsistent with the rest. More discussion is needed for when this is needed and when it is not.
>
> Thank you for your comment. Similar to the ablation studies and as described in the paper, we did not use super-resolution for large datasets such as FSC-147 and Emoji-Count for two simple reasons:
>
> Firstly, and most importantly, the API cost is at least 4× higher for super-resolution images (we use 4× SR on the images).
> Secondly, the images in these datasets retain good resolution after division, and the performance does not change significantly by using or not using super-resolution.
>
> However, to preserve consistency, and despite the extra cost, we experimented with LVLM-Count using GPT-4o both with super-resolution and without super-resolution for all three tables in the manuscript:
>
> Table [1](https://anonymous.4open.science/r/lvlm-count-paper-65B1/tables_updated/table_1_updated.jpg), [2](https://anonymous.4open.science/r/lvlm-count-paper-65B1/tables_updated/table_2_updated.jpg), and [3](https://anonymous.4open.science/r/lvlm-count-paper-65B1/tables_updated/table_3_updated.jpg) (Please click on the numbers to see the corresponding updated tables). We clearly denote the version using super-resolution by "w SR" (with super-resolution) and the version not using the super-resolution by "w/o SR" (without super-resolution) in the updated tables.
>
> The summary is that on the FSC-147 and Emoji-Count datasets, we significantly outperform the base GPT-4o regardless of using super-resolution. However, as mentioned before, in the case of the TallyQA simple benchmark, the performance without super-resolution is slightly lower compared to the base GPT-4o, simply because the generated sub-images lack sufficient quality.
>
> For the reviewers' convenience, we have repeated each table below:
>
> **FSC-147, [Table 1](https://anonymous.4open.science/r/lvlm-count-paper-65B1/tables_updated/table_1_updated.jpg)**:
>
> | Method | MAE | RMSE |
> |---------------------------------------------------------|-------|--------|
> | TFOC | 24.79 | 137.15 |
> | VLCounter | 17.05 | 106.16 |
> | CounTX | 15.88 | 106.29 |
> | DAVEprm | 14.90 | 103.42 |
> | CountGD | 14.76 | 120.42 |
> | GroundingREC | **10.12** | 107.19 |
> | -----------------------------------------------------| ------| --------|
> | Number of target segmentation masks (LVLM+GroundingDINO)| 44.14 | 154.39 |
> | Base GPT-4o counting the target segmentation masks | 38.45 | 38.45 |
> | GPT-4o | 23.75 | 137.39 |
> | LVLM-Count (GPT-4o as LVLM, Cluster-based on both axes, w/o SR) | 17.67 | 90.61 |
> | LVLM-Count (GPT-4o as LVLM, with super-resolution) | 16.76 | 89.60 |
> | LVLM-Count (GPT-4o as LVLM, w/o super-resolution) | 16.23 | **76.09** |
> | Gemini 1.5 Pro | 25.20 | 108.76 |
> | LVLM-Count (Gemini 1.5 Pro as LVLM, w/o SR) | 14.85 | 85.60 |
> | Qwen2 VL 72B AWQ | 33.45 | 145.90 |
> | LVLM-Count (Qwen2 VL 72B AWQ as LVLM, w/o SR) | 17.77 | 113.06 |

---

> ### Author Response · Authors · 2024-12-02
> **Reply to Reviewer 4Qwc Additional  Comments (3/4)**
>
> **TallyQA Simple Benchmark, [Table 2](https://anonymous.4open.science/r/lvlm-count-paper-65B1/tables_updated/table_2_updated.jpg):**
>
> | Method | EA | MAE | RMSE |
> |----------------------------------------------------------------|-------|-------|-------|
> | TFOC | 6.45 | 8.28 | 14.79 |
> | DAVEprm | 5.81 | 16.06 | 35.26 |
> | GroundingREC | 23.87 | 2.90 | 4.71 |
> | CountGD | 41.94 | 2.37 | 4.56 |
> |-------------------------------------------|------|------| -------
> | Number of the target segmentation masks (LVLM+GroundingDINO)| 27.31 | 2.60 | 4.10 |
> | Base GPT-4o counting the target segmentation masks | 25.38 | 2.82 | 2.82 |
> | GPT-4o | 44.30 | 1.38 | 2.35 |
> | LVLM-Count (GPT-4o as LVLM, Cluster-based on both axes, w SR) | 41.29 | 1.82 | 3.33 |
> | LVLM-Count (GPT-4o as LVLM, Cluster-based on x-axis, with SR) | 44.73 | 1.18 | 2.06 |
> | LVLM-Count (GPT-4o as LVLM, Cluster-based on x-axis, w/o SR) | 40.00 | 1.48 | 2.45 |
> | Gemini 1.5 Pro | 47.10 | **1.08** | **1.87** |
> | LVLM-Count (Gemini 1.5 Pro as LVLM, Cluster-based on x-axis, w SR) | 45.16 | 1.43 | 4.72 |
> | Qwen2 VL 72B AWQ | **49.03** | 1.44 | 2.74 |
> | LVLM-Count (Qwen2 VL 72B AWQ as LVLM, Cluster-based on x-axis, w SR) | 41.72 | 1.66 | 3.41 |
>
> **TallyQA Complex Benchmark, [Table 2](https://anonymous.4open.science/r/lvlm-count-paper-65B1/tables_updated/table_2_updated.jpg):**
>
> | Method | EA | MAE | RMSE |
> |----------------------------------------------------------------|-------|-------|-------|
> | TFOC | 1.34 | 12.41 | 22.80 |
> | DAVEprm | 3.36 | 28.36 | 57.56 |
> | GroundingREC | 16.78 | 5.83 | 10.13 |
> | CountGD | 5.37 | 9.78 | 17.21 |
> | ------------------------------------------- | ------ | ------ |------- |
> | Number of the target segmentation masks (LVLM+GroundingDINO) | 15.44 | 4.25 | 6.78 |
> | Base GPT-4o counting the target segmentation masks | 11.41 | 4.99 | 4.99 |
> | GPT-4o | 29.08 | 2.60 | 4.74 |
> | LVLM-Count (GPT-4o as LVLM, Cluster-based on both axes, w SR) | 28.41 | 3.18 | 8.91 |
> | LVLM-Count (GPT-4o as LVLM, Cluster-based on x-axis, with SR) | **34.48** | 2.28 | 4.18 |
> | LVLM-Count (GPT-4o as LVLM, Cluster-based on $x$-axis, w/o SR) | 31.10 | 2.42 | 3.91 |
> | Gemini 1.5 Pro | 25.73 | **2.13** | **3.36** |
> | LVLM-Count (Gemini 1.5 Pro as LVLM, Cluster-based on x-axis, w SR) | 26.62 | 2.79 | 4.70 |
> | Qwen2 VL 72B AWQ | 24.61 | 3.21 | 5.35 |
> | LVLM-Count (Qwen2 VL 72B AWQ as LVLM, Cluster-based on x-axis, w SR) | 30.65 | 2.47 | 4.35 |

---

> ### Author Response · Authors · 2024-12-02
> **Reply to Reviewer 4Qwc Additional Comments (4/4)**
>
> **Emoji-Count Benchmark, [Table 3](https://anonymous.4open.science/r/lvlm-count-paper-65B1/tables_updated/table_3_updated.jpg):**
>
> | Method | MAE | RMSE |
> |---------------------------------------------------------|--------|--------|
> | TFOC | 64.64 | 87.45 |
> | DAVEprm | 198.99 | 208.08 |
> | CountGD | 137.93 | 156.80 |
> | GroundingREC | 36.16 | 51.88 |
> |---------------------------------------------------|--------|--------|
> | Number of the target segmentation masks (LVLM+GroundingDINO) | 82.47 | 107.98 |
> | Base GPT-4o counting the target segmentation masks | 107.72 | 162.12 |
> | GPT-4o | 22.51 | 35.94 |
> | LVLM-Count (GPT-4o as LVLM, Cluster-based on both axes, w/o SR) | 11.10 | 24.23 |
> | LVLM-Count (GPT-4o as LVLM, with Super-Resolution) | **7.31** | **15.97** |
> | LVLM-Count (GPT-4o as LVLM, w/o Super-Resolution) | 9.82 | 19.46 |
> | Gemini 1.5 Pro | 18.17 | 27.83 |
> | LVLM-Count (Gemini 1.5 Pro as LVLM, w/o SR) | 14.35 | 23.42 |
> | Qwen2 VL 72B AWQ | 82.41 | 186.32 |
> | LVLM-Count (Qwen2 VL 72B AWQ as LVLM, w/o SR) | 20.67 | 34.48 |
>
> > Which entry corresponds to the baseline of LVLM + GroundingDINO: After target segmentation step (Sec 3.2), directly counting the masks by SAM?
>
> Thank you for your comment.
>
> Please note that the name of this baseline in Tables [1](https://anonymous.4open.science/r/lvlm-count-paper-65B1/tables_updated/table_1_updated.jpg), [2](https://anonymous.4open.science/r/lvlm-count-paper-65B1/tables_updated/table_2_updated.jpg), and [3](https://anonymous.4open.science/r/lvlm-count-paper-65B1/tables_updated/table_3_updated.jpg) (please click on the numbers to view) is the text after the arrow below:
>
> **LVLM+ GroundingDINO → *Number of the target segmentation masks***
>
> The SAM masks are simply segmentations of the GroundingDINO detections, so they contain an equal number.
>
>
> > How about inputing the LVLM with an image only keeping the SAM mask areas while removing the rest? What results could be obtained?
>
> Thank you for your comment.
>
> Please note that this baseline appears in all three tables (Tables [1](https://anonymous.4open.science/r/lvlm-count-paper-65B1/tables_updated/table_1_updated.jpg), [2](https://anonymous.4open.science/r/lvlm-count-paper-65B1/tables_updated/table_2_updated.jpg), and [3](https://anonymous.4open.science/r/lvlm-count-paper-65B1/tables_updated/table_3_updated.jpg)). You can view them by clicking on the respective numbers. The baseline appears by the following name (the name after the arrow is how it appears in the tables):
>
> **Inputting the LVLM with an image while keeping only the SAM mask areas and removing the rest --> *Base GPT-4o counting the target segmentation masks***
>
> Please note that we outperform these baselines significantly.
>
> >Overall the current ablation study is not convincing yet, there is a need to be more carefully conducted, taking into account the above comments. Thanks.
>
> Thank you for raising your concern. We have updated [Table 4](https://anonymous.4open.science/r/lvlm-count-paper-65B1/tables_updated/table_4_updated.jpg) (click to view) to now show results based on the entire FSC-147 dataset.
>
> ------
> Finally, we sincerely thank you for your comments, which have significantly helped us improve the quality and presentation of our work. We have addressed every comment provided and welcome any further feedback. We sincerely hope that the clarifications and additional results address your concerns effectively. We kindly request you to consider the possibility of revisiting our score.

---

> > ### Author Response · Authors · 2024-12-03
> >
> > We hope the clarifications and additional results provided are sufficient for you to kindly consider improving our score. In any case, we would be grateful if the reviewer could let us know if they have any other concerns which we could answer before the deadline.

---

> > > ### Author Response · Authors · 2024-12-04
> > >
> > > Dear Reviewer,
> > >
> > > We sincerely apologize for reaching out to you once again. We would like to assure you that we have devoted considerable time and effort to addressing each of your comments with thorough explanations and comprehensive experimental results.
> > >
> > > We deeply value your time and expertise and sincerely hope that you will continue to review our responses and share your invaluable feedback. Although the discussion period is nearing its conclusion and further exchanges between reviewers and authors will no longer be possible, we would like to kindly note that the system still allows esteemed reviewers to update their evaluations. We hope that our responses meet your expectations and that you might consider revisiting your evaluation based on the clarifications and improvements provided.
> > >
> > > Thank you once again for your time and effort.
> > >
> > > Warm regards,
> > >
> > > The Authors

---

### Official Review · Reviewer_D4FJ · 2024-11-04

**Soundness:** 2
**Presentation:** 3
**Contribution:** 2
**Rating:** 6
**Confidence:** 5

**Summary:**

This paper aims to improve the counting ability of large VLMs. The paper proposes to split a counting task into sub-tasks by dividing the input image into smaller parts, counting objects in the smaller parts, and then aggregating the counts. The authors show that the proposed approach can generalize to unseen datasets.

**Strengths:**

The results obtained and demonstrated in the paper seem strong.

**Weaknesses:**

The main weaknesses of the paper lie in the lack of enough support for the claims made. In particular, the authors should address the following questions/comments in their responses and revisions:

1. In several places in the paper (e.g. lines 61-62), the authors mention that pipeline detects "the objects of interest". Are there even more than one types of objects to be counted in these datasets? If yes, how are objects of different categories handled? All the visual examples in the paper involve only a single type of object.

2. In line 349, the authors talk about "simple" and "complex" counting questions. What do these mean in this context?

3. Lines 241-242 say "In out experiments, we specify which approach we use for each dataset". Having different approaches for different datasets (outside of a few hyper-parameters) defeats the point of proposing a single approach for a problem. This is problematic, particularly because one of the selling-points of this paper is the claim that the proposed approach generalizes across datasets.

4. The introduction mentions "industry, healthcare, and environmetal monitoring" as the areas of application. It would have been useful if the paper actually included some real-world examples from these domains to demonstrate the utility of the proposed approach.

**Questions:**

Please see the weaknesses section.

Edit: Updated rating from 3 to 6 after reading the authors' responses and updated manuscript.

---

> ### Author Response · Authors · 2024-11-29
> **Response to Reviewer D4FJ (1/3)**
>
> We sincerely thank the reviewer for their time and insights on our manuscript. Below, we address the reviewer’s questions and concerns one by one. Please let us know if you have more questions or concerns. We are happy to discuss them.
>
>
>
> We appreciate your concern regarding the lack of certain experiments and visual examples in the submitted manuscript. We took this feedback seriously and made significant efforts to strengthen the experimental aspect of the paper based on your comments. Accordingly, in the revised manuscript, we have included a substantial number of additional baselines, experiments on new datasets, additional ablation studies, and new visual examples illustrating the performance of LVLM-Count on various benchmarks and real-world applications.
>
>
>
> For quick access to revisions specific to your comments, we have highlighted them in **magenta**. Furthermore, please note that the figure and table numbers in our responses correspond to those in the revised manuscript. For your convenience, we have embedded an anonymized link in the title of each figure/table in our replies, allowing you to view them by simply clicking on the title.
>
>
>
> ----------
>
>
>
> >In several places in the paper (e.g. lines 61-62), the authors mention that pipeline detects "the objects of interest". Are there even more than one types of objects to be counted in these datasets? If yes, how are objects of different categories handled? All the visual examples in the paper involve only a single type of object.
>
>
>
> Thank you for your comment. We break down our response into two parts:
>
>
>
> **Are there multiple types of objects to be counted in these datasets?**
>
> Yes, other than FSC-147 ([Table 1](https://anonymous.4open.science/r/lvlm-count-paper-65B1/tables/table_1.jpg), [Figure 23](https://anonymous.4open.science/r/lvlm-count-paper-65B1/Images/figure_23.jpg)), where most images contain only a single object type, the other two benchmarks, namely TallyQA benchmark ([Table 2](https://anonymous.4open.science/r/lvlm-count-paper-65B1/tables/table_2.jpg)) and Emoji-Count benchmark ([Table 3](https://anonymous.4open.science/r/lvlm-count-paper-65B1/tables/table_3.jpg)) in the manuscript almost always feature images containing more than one object category. Moreover, "Objects of interest" refers to the categories of objects to be counted, which are specified through prompts. One key strength of our method is its ability to generalize well across scenarios involving multiple categories, a challenge for prior trained or training-free models. This superior generalization stems from our model's ability to accurately distinguish between multiple categories using prompts.
>
>
>
> **How are objects of different categories handled?**
>
> We appreciate you highlighting that our manuscript lacked visual examples of a scenario wher multiple object categories are present in the image.
>
>
>
> To address this, we have added Section F in the appendix, which provides visual example specific to this scenario, [Figure 14](https://anonymous.4open.science/r/lvlm-count-paper-65B1/Images/figure_14.jpg) (please click to view). The example shows an image containing three categories: persons, cows, and horses. The pipeline works as follows:
>
>
>
> -   **Counting one of the categories**: In the first row of Figure 14, the question is “How many cows are in the image?” This question is passed to LVLM to extract the "object of interest" (in this case, cows). The “cow” prompt is then sent to the detection model to locate areas containing cows. The segmentation models use the same prompt to find cow masks. Please note that false positive masks don’t cause a problem since we use masks only for object-aware division and not the counting. The generated masks ensure object-aware image division without cutting through cows. The LVLM is then asked, "How many cows are in the image?" for each sub-image. The same process is repeated for counting persons (middle row).
>
>
>
> -   **Counting multiple categories combined**:  Although a challenging scenario, the bottom row of Figure 14 demonstrates how LVLM-Count successfully counts multiple categories combined. The question is “How many cows and persons are in the image?” The LVLM extracts “cow and person” as the object of interest. The detection and segmentation models process areas and masks corresponding to this prompt. The object-aware division happens with respect to all the generated masks. and the LVLM then counts the objects in each sub-image.
>
>
>
>
>
> We sincerely thank you for allowing us to clarify this and apologize for any prior ambiguity.
>
>
>
> ----------

---

> ### Author Response · Authors · 2024-11-29
> **Response to Reviewer D4FJ (2/3)**
>
> >In line 349, the authors talk about "simple" and "complex" counting questions. What do these mean in this context?
>
>
> Thank you for raising this concern. We apologize for forgetting to refer the reader to Appendix C for more details in the main body of the manuscript.
>
>
>
> In the revised manuscript, we have added a sentence on lines 345–346 to guide readers to Appendix C and [Figure 9](https://anonymous.4open.science/r/lvlm-count-paper-65B1/Images/figure_9.jpg). This appendix now provides a detailed explanation of the criteria for simple/complex classification.
>
>
>
> We also briefly discuss simple/complex classification below:
>
> We adopt the terminology from the TallyQA paper (Ref [1]), which introduces the dataset we used to construct the Open-ended Counting Benchmark in our study. TallyQA defines "simple" and "complex" questions as follows:
>
>  “Our simple-complex classifier is made from a set of linguistic rules. First, any substrings such as ‘...in the photo?’ or ‘...in the image?’ were removed from the question. Then, we used SpaCy for part-of-speech tagging on the remaining substring. It was classified as simple if it had only one noun, no adverbs, and no adjectives; otherwise, it was deemed complex. This will classify questions such as ‘How many dogs?’ as simple and ‘How many brown dogs?’ as complex.”
>
> We appreciate this comment, as it helped us clarify this point in our manuscript.
>
>
>
> ----------
>
>
> >Lines 241-242 say "In out experiments, we specify which approach we use for each dataset". Having different approaches for different datasets (outside of a few hyper-parameters) defeats the point of proposing a single approach for a problem. This is problematic, particularly because one of the selling-points of this paper is the claim that the proposed approach generalizes across datasets.
>
>
>
> Thank you for raising this concern about the universality of our method.
>
>
>
> Your valid observation motivated us to run new experiments on the three benchmarks in our work using a universal approach for setting the start and end points of the division paths, which requires no prior information about the dataset. In these experiments, GPT-4o serves as the LVLM in the pipeline, and a cluster-based approach is used along both axes to automatically determine the start and end points of the division paths without relying on prior information about the dataset's object distribution.
>
>
>
> These experiments are referred to as “LVLM-Count (GPT-4o as LVLM, Cluster-based on both axes)” in Tables [1](https://anonymous.4open.science/r/lvlm-count-paper-65B1/tables/table_1.jpg), [2](https://anonymous.4open.science/r/lvlm-count-paper-65B1/tables/table_2.jpg), and [3](https://anonymous.4open.science/r/lvlm-count-paper-65B1/tables/table_3.jpg).
>
>
>
> The brief conclusion from these experiments is that LVLM-Count remains highly effective, even when using the same approach across all datasets. Below is a more detailed explanation:
>
> 1.  **FSC-147 ([Table 1](https://anonymous.4open.science/r/lvlm-count-paper-65B1/tables/table_1.jpg)):**
>
>     “LVLM-Count (GPT-4o as LVLM, Cluster-based on both axes)” still outperforms the base GPT-4o and the prior training-free method, TFOC, by a significant margin.
>
> 2.  **Open-ended counting benchmark ([Table 2](https://anonymous.4open.science/r/lvlm-count-paper-65B1/tables/table_2.jpg)):**
>
>     “LVLM-Count (GPT-4o as LVLM, Cluster-based on both axes)” surpasses all prior trained and training-free methods by a large margin.
>
> 3.  **Emoji-Count ([Table 3](https://anonymous.4open.science/r/lvlm-count-paper-65B1/tables/table_3.jpg)):**
>
>     “LVLM-Count (GPT-4o as LVLM, Cluster-based on both axes)” significantly outperforms the base LVLM and all prior trained and training-free models.
>
> The only case where “LVLM-Count (GPT-4o as LVLM, Cluster-based on both axes)” slightly underperforms the base GPT-4o is in the Open-ended counting benchmark (Table 2). However, this is not because LVLM-Count or the cluster-based approach along both axes lacks robustness. Upon closer inspection, there are other entries in Table 2, such as “LVLM-Count (Gemini 1.5 Pro as LVLM, Cluster-based on the _x_-axis),” also underperform their respective base models despite using prior knowledge (e.g., avoiding horizontal division lines). This is due to the average ground truth value for this benchmark being approximately 7—a range where base LVLMs already perform exceptionally well, making further improvement challenging. This phenomenon can be seen in the example of counting blue circles in [Figure 2](https://anonymous.4open.science/r/lvlm-count-paper-65B1/Images/figure_2.jpg) and in the newly added analysis of performance over different ground-truth intervals of the FSC-147 dataset ([Figure 18](https://anonymous.4open.science/r/lvlm-count-paper-65B1/Images/figure_18.jpg), Appendix H). Figure 18 shows that base LVLMs outperform LVLM-Count for small ground truth values (approximately numbers less than 20).
>
> ----------

---

> > ### Author Response · Authors · 2024-11-29
> > **Response to Reviewer D4FJ (3/3)**
> >
> > >The introduction mentions "industry, healthcare, and environmetal monitoring" as the areas of application. It would have been useful if the paper actually included some real-world examples from these domains to demonstrate the utility of the proposed approach.
> >
> >
> >
> > Thank you for this excellent suggestion.
> >
> >
> >
> > In response, we have added Section G in the appendix, which provides real-world examples:
> >
> >
> >
> > -   **Biology/healthcare**: Bacterial cell counting (Ref [2]) and bone marrow nuclei counting (Ref [3]) added in [Figure 15](https://anonymous.4open.science/r/lvlm-count-paper-65B1/Images/figure_15.jpg).
> >
> > -   **Industry**: Barrel and tree log counting in [Figure 16](https://anonymous.4open.science/r/lvlm-count-paper-65B1/Images/figure_16.jpg).
> >
> > -   **Environmental monitoring**: Penguin counting in studies for penguin population control in the Arctic (Ref [4]) in [Figure 17](https://anonymous.4open.science/r/lvlm-count-paper-65B1/Images/figure_17.jpg).
> >
> >
> >
> > These examples highlight the practicality of our method for challenging real-world scenarios and the improvements it offers over state-of-the-art LVLMs like GPT-4o.
> >
> >
> >
> > ----------
> >
> >
> >
> > We sincerely thank you for your detailed feedback, which helped us address shortcomings in our manuscript. We hope our responses and additional results will be sufficient to kindly consider improving our score.
> >
> > #
> > ------------------
> > References:
> >
> > Ref [1] Manoj Acharya, Kushal Kafle, and Christopher Kanan. TallyQA: Answering complex counting questions. In Proceedings of the AAAI Conference on Artificial Intelligence, volume 33, pp. 8076–8084, 2019.
> >
> >
> >
> > Ref [2] Victor Lempitsky and Andrew Zisserman. Learning to count objects in images. Advances in Neural Information Processing Systems, 23, 2010.
> >
> >
> >
> > Ref [3] Philipp Kainz, Martin Urschler, Samuel Schulter, Paul Wohlhart, and Vincent Lepetit. You should use regression to detect cells. In Medical Image Computing and Computer-Assisted Intervention–MICCAI 2015: 18th International Conference, Munich, Germany, October 5-9, 2015, Proceedings, Part III 18, pp. 276–283. Springer, 2015.
> >
> >
> >
> > Ref [4] Penguin Research. Penguin research webpage, 2016. URL [https://www.robots.ox.ac](https://www.robots.ox.ac/). uk/~vgg/data/penguins/. Accessed: 2024-11-23.

---

> ### Author Response · Authors · 2024-12-01
>
> Hi,
>
> We put significant effort to address all your concerns and we did all additional experiments. Since the end of the author/review period is tomorrow, we would be grateful if the reviewer could reply to the rebuttal.

---

> > ### Author Response · Authors · 2024-12-02
> >
> > We hope the clarifications and additional results provided are sufficient for you to kindly consider improving our score. In any case, we would be grateful if the reviewer could let us know if they have any other concerns which we could answer before the deadline.

---

> > > ### Comment · Reviewer_D4FJ · 2024-12-03
> > > **Thank you**
> > >
> > > I thank the authors for the detailed responses. I believe that the additional information and experiments have satisfactorily answered my questions. I am willing my rating. However, I still have some concern about so much of the important information in the paper being in appendices instead of the main body of the paper within the allocated page limit.

---

> > > > ### Author Response · Authors · 2024-12-03
> > > >
> > > > We would like to express our gratitude to the reviewer for raising their score. We will also take into account your additional comments for our paper. Thank you.

---

### Official Review · Reviewer_ATzA · 2024-11-05

**Soundness:** 3
**Presentation:** 2
**Contribution:** 3
**Rating:** 5
**Confidence:** 4

**Summary:**

This paper addresses the problem of counting objects in images using large vision-language models (LVLMs), which often struggle with counting tasks, particularly when the object count exceeds typical values encountered during training. The authors propose a method named LVLM-Count, which aims to enhance LVLMs' counting abilities through a divide-and-conquer approach. LVLM-Count is structured in four stages: (1) Area Detection, where regions containing relevant objects are identified; (2) Target Segmentation, in which these regions are segmented to highlight individual objects; (3) Object-aware Division, where regions are divided into sub-images without cutting through the segmented objects; and (4) Counting Aggregation, where the LVLM counts objects in each sub-image and combines the results to produce the final count. The proposed approach is claimed to generalize well to new datasets without additional training or fine-tuning, showing improved performance on various datasets and benchmarks compared to prior methods.

**Strengths:**

This paper explores a relatively novel approach by focusing on enhancing counting capabilities in large vision-language models (LVLMs) using a training-free methodology. By leveraging the power of LVLMs, the authors propose an effective paradigm that does not rely on additional training or fine-tuning, which is particularly advantageous in scenarios where labeled data is limited or unavailable. The method demonstrates a creative approach to addressing challenges in object counting, especially in cases with a high number of objects and significant object overlap. By employing a divide-and-conquer strategy, the proposed LVLM-Count effectively manages the complexity of densely populated scenes, providing a practical and scalable solution for counting tasks that would typically challenge standard vision models. The paper’s emphasis on a training-free framework in conjunction with LVLMs is both innovative and valuable, offering a flexible counting solution that could be adapted to various applications without the need for retraining.

**Weaknesses:**

This paper also presents some limitations, as acknowledged in the final section. The proposed method heavily relies on the accuracy of the initial stages—specifically, object detection and instance segmentation. If either of these stages is inaccurate, it could significantly affect the downstream steps, potentially compromising the overall performance. This dependency raises questions about the robustness of the method on more challenging datasets, especially those with high levels of occlusion or camouflage, where accurate detection and segmentation are inherently more difficult. Additionally, the comparison with existing methods is relatively limited. To strengthen the paper’s persuasiveness, it would be beneficial for the authors to include more comprehensive comparisons with other state-of-the-art counting methods. Lastly, the paper would benefit from further ablation studies, such as an investigation into the impact of each stage in the pipeline. For instance, an ablation study examining the effect of including or excluding the target segmentation stage could provide insights into its significance within the proposed approach. These additions could help validate the robustness and effectiveness of the method across diverse scenarios.

**Questions:**

The questions are in weakness part.

**Details Of Ethics Concerns:**

No more concerns.

---

> ### Author Response · Authors · 2024-11-29
> **Response to Reviewer ATzA (1/3)**
>
> We sincerely thank the reviewer for their time and insights on our manuscript. Below, we address the reviewer’s questions and concerns one by one. Please let us know if you have more questions or concerns. We are happy to discuss them.
>
> Please note that in the revised manuscript, we have included a substantial number of additional baselines, experiments on new datasets, additional ablation studies, and additional visual examples showcasing the performance of LVLM-Count on different benchmarks and real-world applications. It is important to note that we performed all experiments that the reviewer suggested.
>
> For quick access to revisions specific to your comments and concerns, we have highlighted the changes in “cyan.” Additionally, please note that the figure and table numbers in our replies correspond to those in the revised manuscript. For your convenience, we have embedded links (anonymized) in the titles of each figure/table mentioned in our replies, allowing you to view them simply by clicking on the title.
>
> > 1.  This paper also presents some limitations, as acknowledged in the final section. The proposed method heavily relies on the accuracy of the initial stages—specifically, object detection and instance segmentation. If either of these stages is inaccurate, it could significantly affect the downstream steps, potentially compromising the overall performance. This dependency raises questions about the robustness of the method on more challenging datasets, especially those with high levels of occlusion or camouflage, where accurate detection and segmentation are inherently more difficult.
>
> We thank you for your suggestion to find a dataset that includes occluded or camouflaged objects to investigate the potential limitations of our method in its initial stages.
>
> We searched the counting literature and identified the penguin dataset Ref [1]. This challenging dataset consistently features heavy occlusion and complex background patterns that can easily be mistaken for penguins Ref [2]. Please see Appendix E in the revised manuscript for more details about this dataset.
>
> We ran our method using GPT-4o, Gemini 1.5 Pro, and Qwen2 VL 72B AWQ Ref [3] as LVLMs on this dataset. The dataset has two splits, mixed-sites split, and separated-sites split. The results for both splits are included in Tables [5](https://anonymous.4open.science/r/lvlm-count-paper-65B1/tables/table_5.jpg) and [6](https://anonymous.4open.science/r/lvlm-count-paper-65B1/tables/table_6.jpg) (please click to view) in Appendix E. A summary of the observations from these tables is as follows: as mentioned in the manuscript, if we set the detection thresholds for the area detection and target segmentation stages to very low values, the limitations in these stages can be alleviated significantly. While this may result in false positive detections, unlike failures to detect, LVLM-Count is fully robust to false positives. This is because the counting is performed at the end by an LVLM, which has greater accuracy and reasoning power.
>
> Additionally, [Figure 11](https://anonymous.4open.science/r/lvlm-count-paper-65B1/Images/figure_11.jpg) (please click to view) visually demonstrates the limitation of the target segmentation stage for a challenging image from the penguin dataset when the threshold is not set to a low value. [Figure 12](https://anonymous.4open.science/r/lvlm-count-paper-65B1/Images/figure_12.jpg) (please click to view) shows how this limitation can be mitigated by lowering the threshold. Similarly, [Figure 13](https://anonymous.4open.science/r/lvlm-count-paper-65B1/Images/figure_13.jpg) (please click to view) illustrates the limitation of the area detection stage for another challenging image from the penguin dataset and how it can also be resolved by setting the detection threshold to a low value.
>
> Below, we provide a simple technical explanation of how we address this limitation in our method:
>
> As described in the manuscript, we use GroundingDino for both the area detection and target segmentation stages. GroundingDino is a state-of-the-art object detector. Using such a powerful model already mitigates this limitation to a certain degree. However, like all object detectors, it includes a hyperparameter—commonly referred to as the _box threshold_ or _detection threshold_—that determines the probability beyond which an object is detected as a target. A key solution we propose in this paper to alleviate this drawback is to set this threshold to a very low value. Lowering the threshold reduces the chances of missing areas or objects of interest. While this approach does increase the number of false positives, an advantage of our method’s design is that false positives do not impact the final count since counting is not performed at this stage. The only effect is that when a division line is drawn, it avoids cutting through these false positive cases.

---

> ### Author Response · Authors · 2024-11-29
> **Response to Reviewer ATzA (2/3)**
>
> >2. Additionally, the comparison with existing methods is relatively limited. To strengthen the paper’s persuasiveness, it would be beneficial for the authors to include more comprehensive comparisons with other state-of-the-art counting methods.
>
>
>
> We highly value your suggestion to extend our comparisons by including even more methods.
>
>
>
> In the submitted version, for the FSC-147 dataset, we included the highest-performing prompt-based models available in the literature. To the best of our knowledge, there are no higher-performing prompt-based models on this dataset.
>
>
>
> However, we agree with your suggestion and appreciate the opportunity to strengthen our position by including additional prior methods, even if they might not outperform the models already included.
>
>
>
> To this end, we have made the following updates to the tables in the manuscript:
>
>
>
> **[Table 1](https://anonymous.4open.science/r/lvlm-count-paper-65B1/tables/table_1.jpg):** We have added VLCounter (Ref [4]) to this table. Additionally, we included results for new base LVLMs: Gemini 1.5 Pro, Qwen2 VL 72B AWQ, and LVLM-Count, which uses these LVLMs.
>
>
>
> **Tables [2](https://anonymous.4open.science/r/lvlm-count-paper-65B1/tables/table_2.jpg) and [3](https://anonymous.4open.science/r/lvlm-count-paper-65B1/tables/table_3.jpg):** We have added DAVE_prm (Ref. [5]) to both tables. Results from experiments with Gemini 1.5 Pro, Qwen2 VL 72B AWQ, and LVLM-Count using these models have also been added.
>
>
>
> **Explanation of Prior Model Selection for Tables 2 and 3**
>
> For experiments such as the Open-ended Counting Benchmark (Table 2) and Emoji-Count (Table 3), we initially included two state-of-the-art trained models, CountGD and GroundingREC, for comparison. These were the top-performing trained models on FSC-147. In response to your suggestion, we have now included the third-highest-performing model on FSC-147, DAVE_prm, to our comparisons.
>
> After adding these new models for comparison, the performance rankings of our methods remain largely unchanged.
>
> -   **[Table 1](https://anonymous.4open.science/r/lvlm-count-paper-65B1/tables/table_1.jpg) (click to view):** Our method continues to outperform all training-free models and base LVLMs by a large margin.
>
> -   **[Table 2](https://anonymous.4open.science/r/lvlm-count-paper-65B1/tables/table_2.jpg) (click to view):** Our method consistently outperforms all trained models and achieves the highest exact accuracy (EA) for complex questions among all models, including base LVLMs.
>
> -   **[Table 3](https://anonymous.4open.science/r/lvlm-count-paper-65B1/tables/table_3.jpg) (click to view):** Our method remains the best among both trained and training-free models as well as the base LVLMs.
>
> The most interesting observation is the performance of Qwen2 VL 72B AWQ. Despite being the best-performing open-source LVLM at the time of our experiments, it is clearly weaker than commercial models like GPT-4o and Gemini 1.5 Pro and performs much worse on its own. However, when incorporated into our pipeline (LVLM-Count), it achieves performance levels comparable to those of GPT-4o and Gemini 1.5 Pro. This highlights how our method significantly improves the counting ability of open-source models, even more so than for commercial ones.
>
> We sincerely thank you for your comment, which has helped us broaden our comparisons.

---

> ### Author Response · Authors · 2024-11-29
> **Response to Reviewer ATzA (3/3)**
>
> >3. Lastly, the paper would benefit from further ablation studies, such as an investigation into the impact of each stage in the pipeline. For instance, an ablation study examining the effect of including or excluding the target segmentation stage could provide insights into its significance within the proposed approach. These additions could help validate the robustness and effectiveness of the method across diverse scenarios.
>
>
>
> Thank you for the suggestion to include more variation in our ablation studies. This will greatly benefit potential readers by providing a clearer understanding of the effectiveness of each stage in our pipeline.
>
>
>
> In the original ablation study in Appendix A, Table 4, we investigated the following:
>
> 1) A pipeline that uses only object-aware division (including target segmentation).
>
> 2) A pipeline that uses area detection + object-aware division (this is the best pipeline and corresponds to our method, **LVLM-Count**).
>
>
>
> Based on your suggestion, in the revised manuscript, we have added other variations to [Table 4](https://anonymous.4open.science/r/lvlm-count-paper-65B1/tables/table_4.jpg) (Please click to view) in Appendix A, which include but are not limited to:
>
>
>
> 1.  A pipeline that uses only area detection.
>
> 2.  A pipeline where both area detection and target segmentation are excluded, and the images are divided by straight lines. We refer to this approach as **Naive Division**.
>
> 3.  A pipeline where area detection is included, but target segmentation is excluded, and images are divided using naive division.
>
>
>
> Moreover, we repeated all the above experiments for a new LVLM, Gemini 1.5 Pro. We hope these additional variations provide more insights into the effects of each stage in the pipeline.
>
>
>
> The ablation results show that the highest-performing version is when both area detection and object-aware division (including target segmentation) are part of the pipeline. This corresponds to our proposed method in the manuscript, **LVLM-Count**.
>
>
>
>
> Finally, we sincerely thank you for your comments, which have helped us greatly to improve the quality of our work and its presentation. We welcome any further feedback. We hope the clarifications and additional results provided are sufficient for you to kindly consider improving our score.
>
> #
> ----------
> References
>
> Ref[1] Penguin Research. Penguin research webpage, 2016. URL [https://www.robots.ox.ac](https://www.robots.ox.ac/).uk/~vgg/data/penguins/. Accessed: 2024-11-23
>
> Ref[2] Carlos Arteta, Victor Lempitsky, and Andrew Zisserman. Counting in the wild. In Computer Vision–ECCV 2016: 14th European Conference, Amsterdam, The Netherlands, October 11–14, 2016, Proceedings, Part VII 14, pp. 483–498. Springer, 2016.
>
> Ref[3]  An Yang, Baosong Yang, Binyuan Hui, Bo Zheng, Bowen Yu, Chang Zhou, Chengpeng  Li, Chengyuan Li, Dayiheng Liu, Fei Huang, et al. Qwen2 technical report. arXiv preprintarXiv:2407.10671, 2024.
>
>
>
> Ref[4] Seunggu Kang, WonJun Moon, Euiyeon Kim, and Jae-Pil Heo. Vlcounter: Text-aware visual representation for zero-shot object counting. In Proceedings of the AAAI Conference on Artificial Intelligence, volume 38, pp. 2714–2722, 2024.
>
>
>
> Ref[5] Jer Pelhan, Vitjan  Zavrtanik, Matej Kristan, et al. DAVE-A: Detect-and-verify paradigm for lowshotcounting. In Proceedings of the IEEE/CVF Conference on Computer Vision and Pattern Recognition, pp. 23293–23302, 2024.

---

> ### Author Response · Authors · 2024-12-01
>
> Hi
>
> We put significant effort to address all your concerns and we did all additional experiments. Since the end of the author/review period is tomorrow, we would be grateful if the reviewer could reply to the rebuttal.

---

> > ### Author Response · Authors · 2024-12-02
> >
> > We hope the clarifications and additional results provided are sufficient for you to kindly consider improving our score. In any case, we would be grateful if the reviewer could let us know if they have any other concerns which we could answer before the deadline.

---

> ### Author Response · Authors · 2024-12-03
>
> Hi,
>
> We apologize for bothering you again about this. The last day that the authors can post a reply is today, and we would be grateful if you could reply to the rebuttal. We hope the clarifications and additional results provided are sufficient for you to kindly consider improving our score. In any case, we would be grateful if the reviewer could let us know if they have any other concerns which we could answer before the deadline.

---

> ### Author Response · Authors · 2024-12-04
>
> Dear Reviewer,
>
> We sincerely apologize for reaching out to you once again. We would like to assure you that we have devoted considerable time and effort to addressing each of your comments with thorough explanations and comprehensive experimental results.
>
> We deeply value your time and expertise and sincerely hope that you will continue to review our responses and share your invaluable feedback. Although the discussion period is nearing its conclusion and further exchanges between reviewers and authors will no longer be possible, we would like to kindly note that the system still allows esteemed reviewers to update their evaluations. We hope that our responses meet your expectations and that you might consider revisiting your evaluation based on the clarifications and improvements provided.
>
> Thank you once again for your time and effort.
>
> Warm regards,
>
> The Authors

---

### Meta-Review · Area_Chair_YSfD · 2024-12-19

**Metareview:**

This paper proposes LVLM-COUNT, a divide-and-conquer framework to enhance the counting ability of large vision-language models (LVLMs) through four stages: area detection, target segmentation, object-aware division, and counting aggregation. Experiments are conducted on multiple datasets.

The main strengths are: 1) the innovative training-free framework with LVLMs, and effectively handles complex scenes with overlapping objects, and 2) good performance.
The main weaknesses are: 1) inconsistent performance (the performance depends heavily on the accuracy of initial object detection and instance segmentation stages), and 2) insufficient experiments, including limited comparison with existing methods, lack of more detailed ablation studies to evaluate each stage of the pipeline, and lack of training and inference efficiency analysis.

After rebuttal, the main issues of inconsistent performance and insufficient ablation studies still remain (recognized by Reviewers 4Qwc, and ATzA), which weakens the contribution of this paper. Thus, the AC does not recommend acceptance at this conference. The authors are encouraged to address these concerns for future submissions.

**Additional Comments On Reviewer Discussion:**

In the initial comments, the main concerns include 1)inconsistent performance (the performance depends heavily on the accuracy of initial object detection and instance segmentation stages), and 2) insufficient experiments, including limited comparison with existing methods, lack of more detailed ablation studies to evaluate each stage of the pipeline, and lack of training and inference efficiency analysis.

In the rebuttal phase, the authors provided more experimental results, and the issues of lack of more detailed ablation studies to evaluate each stage of the pipeline and lack of training and inference efficiency analysis are well addressed. Reviewer D4FJ and wTin increased their scores from 3, 5 to 6, 6.

However, Reviewer 4Qwc considers the provided ablation study is not convincing, and the issues of inconsistent performance also remain.

The final scores are 6, 6, 5, and 3.

---

### Decision · Program_Chairs · 2025-01-22

Reject